# *Tephrosia purpurea*, with (-)-Pseudosemiglabrin as the Major Constituent, Alleviates Severe Acute Pancreatitis-Mediated Acute Lung Injury by Modulating HMGB1 and IL-22

**DOI:** 10.3390/ijms26062572

**Published:** 2025-03-13

**Authors:** Gamal A. Soliman, Mohammed A. Alamri, Rehab F. Abdel-Rahman, Marawan A. Elbaset, Hanan A. Ogaly, Maged S. Abdel-Kader

**Affiliations:** 1Department of Pharmacology and Toxicology, College of Pharmacy, Prince Sattam Bin Abdulaziz University, Al-Kharj 11942, Saudi Arabia; g.soliman@psau.edu.sa (G.A.S.); ma.alamri@psau.edu.sa (M.A.A.); 2Department of Pharmacology, National Research Centre, Giza 12622, Egypt; rf.abdelrahman@nrc.sci.eg (R.F.A.-R.); masayed@iu.edu (M.A.E.); 3Stark Neurosciences Research Institute, Indiana University School of Medicine, Indianapolis, IN 46202, USA; 4Department of Neurology, Indiana University School of Medicine, Indianapolis, IN 46202, USA; 5Department of Biochemistry, College of Veterinary Medicine, Cairo University, Giza 12613, Egypt; hananogaly@cu.edu.eg; 6Department of Pharmacognosy, College of Pharmacy, Prince Sattam Bin Abdulaziz University, Al-Kharj 11942, Saudi Arabia

**Keywords:** pulmonary function, *Tephrosia purpurea*, pseudosemiglabrin, iNOS, pancreas, ischemia, reperfusion

## Abstract

Ischemia-reperfusion (IR) injury is a major cause of multiple organ failure. The purpose of this study was to look into the role of *Tephrosia purpurea* (TEP) and its active constituent pseudosemiglabrin (PS) in alleviating severe acute pancreatitis and its associated acute lung injury. We established a rat pancreatic IR model, and the rats were treated with TEP (200 mg/kg and 400 mg/kg) and PS (20 and 40 mg/kg), in addition to the IR control and sham groups. The results showed that the respiratory parameters, including inspiratory time (Ti), expiratory time (Te), duration (Dr), and respiratory rate (RR), were comparable among all groups, while peak inspiratory flow (PIF), forced vital capacity (FVC), and forced expiratory volume at 0.1 s (FEV_0_._1_) were significantly impaired. Notably, PS at 40 mg/kg showed normal PIF, FVC, and FEV_0_._1_/FVC compared to the IR group, indicating an improved lung function. Additionally, TEP and PS showed protective effects on pancreatic and lung tissues compared to the IR control group, with the following effects: alleviating pathological damage; reducing serum levels of trypsinogen activation peptide (TAP), lipase, and amylase; decreasing oxidative stress markers such as MDA and MPO; restoring antioxidant enzyme activity (GPx); suppressing inflammatory markers TNF-α, IL-6, and NF-κB; downregulating HMGB1 gene in pancreatic tissue; and upregulating the IL-22 gene in lung tissues. In conclusion, the obtained findings demonstrate that oral supplementation of TEP and PS to rats with pancreatic IR alleviates pancreatic and lung injuries by reducing oxidative stress and modulating inflammatory processes, which offers an attractive therapeutic option for severe acute pancreatitis and its associated acute lung injury.

## 1. Introduction

Ischemia is the lack of blood flow to a tissue as a result of constricted or blocked arteries, which leaves the tissue devoid of the nutrients and oxygen needed for cellular metabolism. It is commonly recognized that ischemia, or reduced blood flow, can cause tissue damage and malfunction of the organs. The length and intensity of the ischemia insult dictate the tissue’s ability to recover from the damage and its eventual survival [1]. The vulnerability of various organs to ischemic harm varies. Pancreatic ischemia-reperfusion (IR) injury is a major clinical concern, especially in cases involving pancreatic surgery and organ transplantation [2]. Systemic inflammatory responses resulting from pancreatic ischemia-reperfusion contribute to increased white blood cell counts and the production of oxygen free radicals and release of cytokines [3]. These inflammatory responses can lead to conditions such as acute pancreatitis (AP) where there is an increase in leukocyte activity and the release of pro-inflammatory cytokines, leading to systemic inflammatory response syndrome (SIRS) [4]. Consequently, ischemia-reperfusion can affect other organs in addition to the ischemic organ itself, potentially resulting in multi-system organ failure [5].

The main cause of pancreatic injury is oxidative stress, which can happen in a number of clinical situations, such as heart surgery, hemorrhagic shock, hypothermia, and pancreas transplantation [6]. According to Hoffmann et al. [7] and Rolim et al. [8], the generation of reactive oxygen species (ROS) is able to induce AP which overwhelms the body’s antioxidant defenses by inducing inflammatory cell infiltration, interstitial edema, enzymatic elevation, and pro-inflammatory cytokine activation. Additionally, oxidant chain reaction raises the permeability of the pancreatic microcirculation. This results in microvascular leakage as well as a lack of perfusion, because of the intravascular no-reflow phenomenon brought on by the swelling of endothelial cells, neutrophil adhesion, thromboxane A2, and other vasoactive mediators [9]. This impairment of the microcirculation of the pancreas in the early stage of acute pancreatitis plays an important role in the pathogenesis of this disease [10].

The lungs are generally the most commonly targeted organ in AP, and pulmonary injury resulting from pancreatitis is the leading cause of early death in patients with AP [11]. One of the main causes of AP-induced lung injury is the production of inflammatory mediators, as well as ROS. These mediators aid in the build-up of neutrophils and macrophages, which in turn sets off a series of pathological alterations in the lungs’ pulmonary microcirculation, which eventually result in the development and exacerbation of lung injury caused by AP [12].

*Tephrosia purpurea* (TEP) (Leguminosae) has traditionally been used to treat diabetes mellitus and is thought to be helpful for conditions affecting the kidneys, liver, and spleen [13]. Numerous pharmacological effects, including antioxidant, anti-inflammatory [14], hepato-protective [15], immunomodulatory [16], and anti-helicobacter pylori [17], are exhibited by the plant extracts. According to Gokhale and Saraf [18], phytochemical screening of TEP showed the presence of polyphenols, sterols, retinoids like rotenone, flavonoids like purpurin, and flavonoid glycosides like rutin and osyritin. Pseudosemiglabrin (PS), a flavone derived from TEP [19], has demonstrated anti-inflammatory properties, as evidenced by its impact on interleukin-1 (IL-1), tumor necrosis factor-α (TNF-α), and nitric oxide (NO) levels in vitro. In vivo studies have further supported its anti-inflammatory potential by inhibiting granuloma tissue formation and reducing cytokine production [20,21].

TEP extract appears to be useful in encouraging pancreatic regeneration due to its cytoprotective, anti-inflammatory, and antioxidant properties [22]. Nevertheless, there is a dearth of experimental evidence in this area, so the current study was planned to investigate the possible protective effects of TEP and its active constituent, PS, against AP induced by IR in a rat model.

## 2. Results

### 2.1. Respirometer

The inspiratory time (Ti), expiratory time (Te), duration (D), and respiratory rate (RR) of the IR control, TEP-200, TEP-400, PS-20, and PS-40 groups were comparable to the sham group (Figure 1).

On the contrary, the peak inspiratory flow (PIF), forced vital capacity (FVC), and FEV_0_._1_ were hampered in the IR control compared to the sham group. Similarly, the TEP-200, TEP-400, and PS-20 groups were comparable to the IR control. At the same time, the PS-40 group showed a substantially higher FEV_0_._1_ and PIF value compared to the IR control group, with a comparable value to the sham group (Figure 2).

The IR PEF, FVC, FEV_0_._1_, and FEV_0_._1_/FVC were reduced compared to the sham group. Meanwhile, only rats administered with TEP-400 and PS-40 significantly increased their PEF flow and FEV_0_._1_/FVC compared to the IR control group. Similarly, only PS-40 groups showed an appreciable increment in the FVC ratio compared to the IR control group, as well as comparable values of FEV_0_._1_ to both the IR and sham groups.

### 2.2. Serum Parameters

As shown in Table 1, serum amylase and lipase levels in the IR control group increased more than 4-fold and the TAP level increased more than 7-fold compared to the sham rats. The TEP-200 or PS-20 groups showed a decrease in serum amylase levels by about 36%, while the TEP-400 or PS-40 groups showed a decrease of about 65% compared to the IR control group. Similarly, serum lipase levels decreased by 70% in the TEP-400 or PS-40 groups and by about 40% in the TEP-200 or PS-20 groups compared to the IR control group. Further, treatment with TEP-200, TEP-400, PS-20, or PS-40 reduced TAP in the serum of the rats by 35, 68, 29, or 55% compared to the IR control group, respectively.

### 2.3. Assessment of Inflammatory Markers in Pancreatic and Lung Homogenates

The results showed that the pancreatic and lung TNF-α, IL-1β, and NF-κB contents were much higher in the IR control group than in the sham group (Table 2 and Table 3). The TEP-200 or TEP-400 and PS-20 or PS-40 groups showed a 40, 77, 23, or 69% reduction in pancreatic TNF-α levels compared to the IR control group, respectively. Similarly, administration of TEP-200 or TEP-400 and PS-20 or PS-40 reduced lung TNF-α levels by 59, 82, 46, or 60% compared to the IR control group, respectively. In addition, the pancreatic and lung contents of IL-1β in rats exposed to TEP-200 or TEP-400 and PS-20 or PS-40 were significantly reduced as compared to the IR control group. Moreover, the pancreatic NF-κB in the TEP-200 or TEP-400 and PS-20 or PS-40 groups were reduced by 46, 75, 46, or 75% compared to the IR control group, respectively. At the same time, the TEP-200 or TEP-400 and PS-20 or PS-40 lowered the lung NF-κB by 44, 63, 32, or 62% versus the IR control group, respectively.

### 2.4. Assessment of Pancreatic Oxidative Stress Markers

Pancreatic MDA was elevated 6-fold in the IR control group compared to the sham rats. Pancreatic MDA was reduced by 41, 63, 26, or 55% in the TEP-200 or TEP-400 and PS-20 or PS-40 groups versus the IR control group, respectively (Table 4). At the same time, activities of GPx were reduced by 76.6% and MPO increased by 7.1-fold in the pancreatic tissue of the IR control group compared to the sham rats. TEP-200 or TEP-400 and PS-20 or PS-40 increased GPx by 2.9, 4.5, 2.8, or 3.9-fold compared to the IR control rats, respectively. Meanwhile, pancreatic MPO of TEP-200 or TEP-400 and PS-20 or PS-40 were reduced compared to the IR control group.

### 2.5. Gene Expression Analyses

This study further explored some of the mechanisms by which TEP and PS could regulate the inflammatory response. As detected by RT-qPCR, a marked overexpression in the pancreatic HMGB1 (4.99-fold) was evident in the IR group compared to the sham (Figure 3). Interestingly, treatment with TEP and PS significantly decreased HMGB1 expression to about 2.2, 1.5, 2.8, and 1.9-fold compared to the sham in TEP-200, TEP-400, PS-20, and PS-40 groups, respectively. On the other hand, the obtained results demonstrated that, following pancreatic IR induction, the expression levels of IL-22 in the lung tissues were markedly downregulated to about 0.3-fold compared to the sham group. Meanwhile, administration of TEP-200, TEP-400, PS-20, and PS-40 were efficient to restore the mRNA expression of IL-22 (0.8, 1.2, 0.9, or 1.1-fold, respectively) in the lung compared to the sham group.

### 2.6. Histopathological Examination of Pancreatic Tissues

As represented in Figure 4, the pancreas tissue of the sham group showed a normal histological structure, while pancreatic sections of IR controls showed hemorrhage and edema between pancreatic acini. TEP and PS-treated groups showed congestion and slight edema between pancreatic acini which was dose dependent.

### 2.7. Histopathological Examination of Lung Tissues

Figure 5 depicts the histopathologic investigation of the lung tissue. The IR control group showed the congestion of perialveolar blood vessels with the infiltration of interstitial tissue by mononuclear inflammatory cells. The TEP-treated group showed emphysematous alveoli with the infiltration of interstitial tissue by mononuclear inflammatory cells, while PS-treated groups revealed the infiltration of interstitial tissue by a low number of mononuclear inflammatory cells.

### 2.8. Immunohistochemical Examination of iNOS in Pancreatic Tissue

The immunohistochemical study of the pancreas in the sham group showed a negative reaction for nitric oxide synthase (iNOS) in pancreatic acini (Figure 6). However, a strong positive reaction for iNOS in nuclei of pancreatic acini was detected in the IR control. All treated groups showed a strong reaction for iNOS; however, only a mild reaction was observed in the cytoplasm of some pancreatic acini of the PS-40-treated group.

### 2.9. Immunohistochemical Examination of iNOS in Lung Tissue

The immunohistochemical investigation of the lung tissue of sham group showed a negative reaction for iNOS in pneumocytes (Figure 7). However, a strong positive reaction for iNOS in the cytoplasm of pneumocytes was spotted in the IR control. On the other hand, low doses of TEP and PS-treated groups showed a strong reaction for iNOS, and a very mild reaction was observed in groups treated with high doses of TEP and PS in a few numbers of pneumocytes.

## 3. Discussion

Pancreatic IR injury is a serious clinical concern. The length and intensity of the ischemia insult dictate the tissue’s ability to recover from its harm and eventually its survival [23,24]. The general explanation for ischemic tissue damage is a prolonged, severe tissue hypoxia followed by cellular ATP depletion. The present study investigated the role of TEP and its active ingredient, PS, in protecting the pancreas and lung from IR injury in rats.

In this study, we found that IR following splenic artery clamping resulted in severe histological damage in pancreatic tissue in rats, which is associated with systemic oxidative stress, pro-inflammatory cytokine release, and the loss of cellular integrity in the lung tissue. Exposure of an organ to IR leads to inflammatory activation in other non-ischemic organs, leading to failure of these organs due to elevated levels of oxidative stress and an inflammatory reaction in the re-perfused tissues [25,26]. In the process of severe AP, the lung is the most easily affected organ beside the pancreas [27]. The generation of inflammatory mediators, including cytokines and chemokines, is a fundamental cause of AP-induced lung injury. These mediators contribute to the accumulation of macrophages and neutrophils and, later, trigger a cascade of pathological changes in the microcirculation of the lungs, leading to the occurrence of an AP-induced lung injury [12]. Oxidative stress plays a crucial role in the development of lung injury in acute pancreatitis. An excessive amount of ROS can be generated by activating neutrophils, leading to severe oxidative stress damage [28]. Oxidative stress damage not only plays a role in the local damage of the pancreas but also plays a dominant role in damage to other organs [29].

Evaluating respiratory parameters provides critical insights into the impact of pancreatic IR injury on lung function; it also aids in assessing the treatment efficacy of TEP or PS. In the current work, the respiratory parameters that were selected for analysis were inspiration time (Ti) and expiration time (Te), which are the time elapsed during each breath cycle from inhale to exhale/or from exhale to inhale, respectively. Interestingly, during our study Te, Ti D (duration of respiratory cycle), and RR (i.e., the number of breaths per minute) showed no significant deviations across all groups from their normal values, while other critical measurements changed in an appreciable manner after an IR insult. The fall in peak inspiratory flow (PIF) and peak expiratory flow (PEF), denoting the maximum amount of gas inhaled/or per unit time, respectively, were observed in the IR control group, which implies poor airway function. This hints at a possible presence of bronchial obstruction accompanied by decreased strength in respiratory muscles, such as the diaphragm and intercostals, among others [30]. This alteration has great importance because it reflects a general impact of pancreatic IR on respiratory mechanics (increased alveolar permeability, pulmonary edema, and decreased blood oxygen saturation) and agrees with those seen during acute lung injury in severe pancreatitis [27].

Similarly, a decrease in forced vital capacity (FVC) as well as forced expiratory volume at 0.1 s (FEV_0_._1_) in the IR control group typically indicates the development of lung disease secondary to inflammatory mediated changes of lung compliance and elastic recoil [27,31]. Hence, those levels are prime in appreciating the integration of pulmonary organs, as well as indicating anomalies prevalent in the lung. Treatment with TEP-400 and PS-40 exhibited a potential cure; PS-40 displayed significant improvements on PIF, FVC, and FEV_0_._1_/FVC ratios comparable to the sham group. Restoration of respiratory function was therefore obtained by PS-40, which provides a possible treatment avenue for lung injuries caused by pancreatic IR. Moreover, the dose-dependent manner observed for both TEP and PS implies that the treatment dose has a direct correlation with therapeutic effectiveness. Furthermore, TEP-400 treatment significantly improved the FEV_0_._1_/FVC ratio, implying that the airways are more patent than before, and that the lung is more compliant. This indicates that these drugs could potentially prevent or reverse both constrictive and obstructive diseases involving lung parenchyma with IR injury caused by pancreatic IR.

Amylase and lipase enzymes are synthesized in the pancreatic acinar cells. They are vital enzymes produced by the pancreas and are the most commonly used biochemical markers in the diagnosis of acute pancreatitis [32]. In the present study, serum amylase and lipase levels in the IR control group increased more than 4-fold compared to the sham rats. In this respect, a raised level of serum amylase activity (at least three times the upper limit of normal) supports the diagnosis of acute pancreatitis [33]. Treatment with TEP-400 and PS-40 accelerated the normalization of serum lipase activity. Among the evidence that may support the therapeutic properties of the plant in the course of IR-induced AP is its ability to attenuate serum lipase activity. This enzyme is released by pancreatic acinar cells into the interstitial tissue during AP, and its concentration in the serum serves as an indicator of AP severity with high sensitivity and specificity [34].

Furthermore, the reduction in inflammatory markers such as TNF-α, IL-6, and nuclear factor kappa B (NF-κB) in both the pancreas and lungs of treated groups demonstrates the anti-inflammatory effect of TEP and PS. These findings are consistent with a previous in vitro study that reported the role of PS in reducing the inflammatory markers TNF-α, IL-1, and NO in rat granuloma as well as human macrophages [21].

Small, nonstructural proteins called cytokines are mainly involved in the way the body reacts to injury, illness, and ischemia [35]. A pleiotropic cytokine, TNF-α, can result in tissue damage following tissue ischemia and leukocyte infiltration. TNF-α damages ischemic tissue by altering the permeability of vascular endothelial cells, causing vascular dysfunction, and by interacting with endothelial cells to trigger coagulation events [36].

TNF-α inhibition has been shown to be a potential treatment method to lessen the severity of IR injury. In a rat model of IR-induced lung injury, an anti-TNF-α antibody prevented microvascular damage [5].

IL-1β is a well-known mediator of acute inflammation which plays a crucial role in the release of other members of the pro-inflammatory cytokine cascade, including tumor necrosis factor alpha, platelet-activating factor, prostaglandins, and pro-inflammatory interleukins, and consequently stimulates the development of AP [37,38]. Moreover, it has been demonstrated that early and sustained activation of inflammatory cells with the successive release of IL-1β and other cytokines is responsible for the intense local and systemic inflammatory response in AP [34]. Contrarily, inhibition of the cytokine cascade at the level of the IL-1 receptor, before or soon after induction of pancreatitis, markedly attenuates the rise in these cytokines and is associated with a decreased severity of pancreatitis and reduced pancreatic damage [37,39].

TEP and PS used in our study have been shown to modulate the inflammatory response by reducing the release of pro-inflammatory cytokines in both pancreatic and lung tissues compared to the IR control group. By activating the cytokine cascade and iNOS, IR has been shown to cause severe acute necrotizing pancreatitis with a high mortality rate and to cause a systemic inflammatory response. This implies a connection between the apoptotic process in the lung and pancreas and the excess NO produced by iNOS [40]. Those effects were alleviated upon treatment with TEP and PS, which was made evident by the results of the immunohistochemical examination of both pancreatic and lung tissues in a dose-dependent manner.

In the current study, pancreatic malondialdehyde (MDA) was elevated 6-fold in the IR control group compared to the sham rats. The TEP-200 or TEP-400 and PS-20 or PS-40 groups revealed a significant reduction in pancreatic MDA versus the IR control group. Additionally, activities of glutathione peroxidase (GPx) and myeloperoxidase (MPO) in the pancreatic tissue were reduced in the IR control group compared to the sham rats, and were significantly elevated in TEP-200 or TEP-400 and PS-20 or PS-40 compared to the IR control rats, respectively. The restoration of pancreatic oxidative stress indicators, such as MDA, GPx, and MPO in the therapy groups supports antioxidant capabilities. This is critical in the context of IR injury, because oxidative stress is a major contributor to tissue damage.

ROS and microvascular blood flow failure appear to be important pathological mechanisms of acute pancreatitis caused by IR [41]. Several studies have shown that oxidative stress is the primary mechanism of IR injury in the pancreas [42]. Another main effect of ROS is lipid peroxidation [43]. Lipid peroxidation is brought on by tissue ischemia and reperfusion [44]. It is a series of events that result in the oxidation of polyunsaturated fatty acids. This process damages cellular membranes and generates hazardous byproducts, including MDA [45]. MDA is a useful measure of the degree of lipid peroxidation, which is a free radical-generating mechanism that may be strongly linked to IR-induced tissue damage with an increase in reperfusion [46]. Further, the body has an in-built anti-oxidation system that keeps the production and removal of free radicals in a state of dynamic equilibrium, safeguarding the integrity and functionality of cells. GPx, MPO, and other enzyme systems are among these aforementioned systems [47]. MPO is considered to be a marker of local neutrophil activity causing tissue damage in various inflammatory diseases, including AP [48]. In addition, in patients with AP, it has been shown that MPO contributes to the production of reactive oxygen metabolites and its level depends on the severity of AP, as well as on cytokine blood level [49].

In this study, pancreatic IR resulted in severe oxidative stress, characterized by the increased activity of free radical products, MDA and MPO, as well as decreased GPx activity in the pancreatic tissue of IR control rats. These parameters correlate well with disease severity [50,51]. Administration of TEP-200, TEP-400, PS-20, and PS-40 showed significant protection against the oxidative status of pancreatic tissue of rats, via reducing free radical-derived products (MDA) and increasing the activities of GPx and MPO in comparison to the IR control group. TEP-400 and PS-40 restored GPx activities in the pancreas to levels similar to those in the sham group. These findings are corroborated by the fact that TEP’s antioxidant potential is primarily due to its high polyphenol and flavonoid contents. This is consistent with a previous report that found a substantial link between a plant’s antioxidant potential and its polyphenolic content [52].

Accumulating evidence highlighted the linking interaction between HMGB1 and various inflammatory signaling pathways [53]. HMGB1 is a nuclear protein which belongs to the damage-associated molecular pattern family (DAMP) and acts as a late proinflammatory mediator in tissue injury [54]. Once released, HMGB1 activates toll-like receptors (TLRs) which subsequently stimulate NF-κB signaling [55]. Consistent with previous studies, the present study demonstrated a marked increase in pancreatic HMGB1 expression in the IR group compared with the sham group [56]. In our study, TEP and PS downregulated HMGB1 gene expressions, with a concomitant decrease in the inflammatory mediators NF-κB, TNF-α, and IL-1β in the pancreas and lung. This multipronged mechanism verifies the anti-inflammatory potential of TEP and its PS derivative.

TEP and PS downregulate HMGB1, which is a DAMP that activates Toll-like receptor 4 (TLR4), leading to NF-κB-driven inflammation. HMGB1 release during cellular stress leads to increased pro-inflammatory cytokine production, such as TNF-α and IL-6, as well as neutrophil infiltration, increasing tissue damage [57]. In pancreatic IR, suppression of HMGB1 has been demonstrated to reduce systemic inflammation and organ failure. Studies have shown that natural flavonoids, comparable to TEP polyphenols, reduce HMGB1–TLR4 interactions, preventing NF-κB activation and cytokine production [58]. This suggests that TEP and PS disrupt the DAMP-mediated inflammatory cascade at its initiation.

Evidence has also highlighted the pro-inflammatory and anti-inflammatory roles of Th22 cells in the pathogenesis of inflammatory and autoimmune diseases. Th22 cells are identified as CD4+ cells, where IL-22 primarily mediates their function [59]. IL-22 is one of the IL-10 family members which acts through IL-22 receptor 1 to coordinate immune cells, regulate inflammation, and repair tissue [60]. Previous studies reported that IR-induced activation of L-22/IL-22R1/STAT3 signaling subsequently regulates mitochondrial apoptosis [61]. In addition, IL-22 has been reported to induce Bcl-2, which binds to Beclin-1 and suppresses autophagy [62]. Several animal model studies have reported a significant reduction in IL-22 expression and Th22 cells in lung tissues following acute pancreatitis [63,64]. Consistent with the role of Th22 cells in IL-22 production, our data suggest that TEP and PS may enhance IL-22 levels, potentially mitigating acute lung injury [60].

The upregulation of IL-22 in treated groups may reflect enhanced tissue repair. IL-22 activates STAT3 in alveolar epithelial cells, leading to increased proliferation and mucus production [65]. In pancreatitis-associated lung injury, IL-22 deficiency exacerbates damage, while exogenous IL-22 reduces apoptosis via Bcl-2 induction [66]. TEP and PS may thereby enhance IL-22 signaling, inhibiting IR-induced autophagy and promoting epithelial regeneration.

The regulatory effects of TEP and PS on pancreatic HMGB1, pulmonary IL-22 levels, and Th22 cell-associated markers suggest a plausible explanation for the observed protective benefits. Modification of these genes may contribute to TEP and PS’s anti-inflammatory and tissue-protective properties.

The observed increase in IL-22 levels in TEP/PS-treated groups aligns with the known role of Th22 cells in producing this cytokine. By modulating Th22 cell activity or IL-22 secretion, TEP and PS may promote tissue recovery in IR injury, as IL-22 signaling through STAT3 enhances epithelial regeneration [67].

The histopathological examination further supports the protective effects of TEP and PS treatments on both pancreatic and lung tissues. These observations align with the functional improvements in respiratory parameters and the modulation of inflammatory and oxidative stress markers.

## 4. Materials and Methods

### 4.1. Plant Material

The plants of *Tephrosia purpurea* L. (Pers.) were collected from Gazan in southern Saudi Arabia in November 2016. Details regarding plant identification and the method of extraction, isolation, and identification of (-)-pseudosemiglabrin were recently described [68]. Selection of the examined doses of *T. purpurea* and pseudosemiglabrin was carried out as previously reported [20,69].

### 4.2. Animals

After the approval of the animal experimental protocol by the Ethical Committee for Medical Research, National Research Centre, Egypt, about thirty-six adult Male Wistar rats weighing 180–200 g were purchased from the Animal Facility of the National Research Centre, Egypt. Animals were kept in standard cages, under pathogen-free conditions, and maintained under a controlled room temperature and under normal dark–light cycles. The rats were provided with standard food and water ad libitum, and were allowed to adapt to these conditions for 1 week before beginning the experimental protocol. Experiments were performed according to the National Regulations of Animal Welfare and the Institutional Animal Ethical Committee (IAEC), Approval #2416072022.

### 4.3. Experimental Design

Thirty-six adult male Sprague Dawley rats were randomly divided into 6 equal groups.

Sham group: Rats received 1 mL of the vehicle (2% Tween 80 in sterile saline).IR control group: Rats orally received 1 mL of the vehicle kg.TEP-200 group: Rats treated orally with *T. purpurea* at 200 mg/kg.TEP-400 group: Rats treated orally with *T. purpurea* at 400 mg/kg.PS-20 group: Rats treated orally with pseudosemiglabrin at 20 mg/kg.PS-40 group: Rats treated orally with pseudosemiglabrin at 40 mg/kg.

The vehicle, TP, and PS were given to rats daily for 5 days before the induction of pancreatic ischemia, and once after 24 h of reperfusion.

### 4.4. Induction of Pancreatic Ischemia

One hour after the fifth day dose of the vehicle, TEP extract, or PS, the rats were anesthetized with ketamine (50 mg/kg i.p). A midline abdominal incision was conducted to expose the splenic artery. The splenic artery was occluded with a microvascular clamp for 60 min. Ischemia was confirmed by the blanching of the pancreas. After 60 min, the clamp was removed, and pancreatic reperfusion was observed visually. The skin incision was sutured with silk suture and the rats were allowed to recover [55]. The sham-operated group were subjected to the same conditions but without splenic artery occlusion. Another oral dose of the vehicle, TEP extract, or PS was given to the rats 24 h after the beginning of pancreatic reperfusion.

One hour later, an assessment of the respiratory functions of rats using a respirometer was carried out. Later, blood samples were collected from the retro-orbital venous plexus of rats for estimating serum biochemical parameters. The serum was separated by centrifugation for 1538× *g* for 10 min.

Rats were euthanized by decapitation; the pancreas and lung were separated for histopathological, immunohistochemical, molecular, and biochemical investigations. Tissues of three animals in each group were fixed in 10% formalin buffer for 24 h for histological examination, while the other tissue samples of the rest of animals in each group were dissected and divided into two portions. The first portion was immediately snap-frozen in liquid nitrogen and stored at −80 °C for gene expression analysis, while the second portion was immediately homogenized in ice-cold 10% (*w*/*v*) phosphate buffer. The homogenate was centrifuged at 1800× *g* for 10 min at 4 °C. The supernatant was used for different biochemical analyses.

### 4.5. Assessment of Respiratory Functions Using Respirometer

Inspiratory time (Ti), expiratory time (Te), duration (D), peak inspiratory flow (PIF), peak expiratory flow (PEF), forced vital capacity (FVC), forced expiratory volume 1 (FEV_1_), the ratio of FEV_1_/FVC, and respiratory rate (RR) were estimated by putting the rats in a 1.5 L chamber connected to a spirometer with an MTL1 flow meter and AID instruments^®^ Power Lab gear. After calibrating and acclimatizing for 5 min, labChart 8 was used to analyze the respiratory wave and to detect forced expiration for 15 min.

### 4.6. Serum Analysis

Serum amylase, lipase, and trypsinogen activated peptide (TAP) were analyzed according to the manufacturer’s protocol of the used kit.

### 4.7. Assessment of Inflammatory Markers in Pancreatic and Lung Homogenate

TNF-α, IL-1B, and NF-κB were assessed in lung homogenates using ELISA kits procured from BioVision Inc. (Milpitas, CA, USA).

### 4.8. Assessment of Pancreatic Oxidative Stress Markers

Different oxidative stress markers such as MDA, GPx, and MPO were assessed in pancreatic homogenates using the ELISA technique according to the manufacturer’s protocol (BioVision Inc., Milpitas, CA, USA).

### 4.9. Real-Time Polymerase Chain Reaction (RT PCR)

Total RNA from the pancreas and lung samples was purified with the use of the TRIzol reagent (Thermo Fisher Scientific, Waltham, MA, USA) in accordance with the manufacturer’s protocol. RNA quantity, quality, and integrity were measured with spectrophotometric analysis using the A260/A280 ratio, A260/A230 ratio, and 1% agarose gel electrophoresis. RNA was used as a template in the reverse transcription reaction to form cDNA with the use of the RevertAid First Strand cDNA Synthesis Kit (Fermentas, Thermofsher Scientifc, USA). Then, PCR quantitative analyses of the target gene expression were performed adopting a QuantiTect SYBR Green Kit (Qiagen, Hilden, Germany) on a Real-Time PCR Detection System (Bio-Rad Laboratories, Hercules, CA, USA). The cycling program was as follows: pre-denaturation (94 °C, 5 min), 40 cycles of denaturation (94 °C, 45 s), primer annealing (58 °C, 45 s), primer extension (72 °C, 55 s), and a final extension (72 °C, 7 min). Specific primers for detection of HMGB1 and IL-22 were used: HMGB1, forward 5′-GAGTACCGCCCAAAAATCAA-3′, reverse 5′-TTCATCCTCCTCGTCGTCTT-3′ (XM_039100280.1); IL-22, forward 5′-AGTCACCTCAGCCCCTGTCAC-3′, reverse 5′-CCCGATCGCTTTAATCTCTCC-3′ (NM_001191988.1); and GAPDH, forward 5′-CCTCGTCTCATAGACAAGATGGT-3′, and reverse 5′-GGGTAGAGTCATACTGGAACATG-3′ (XM_039082880.1). The relative expression of target genes was normalized to GAPDH as an endogenous control gene, calculated using the 2^−ΔΔCt^ formula, and expressed as a fold change of the normal control group [70].

### 4.10. Histopathological Examination of Pancreatic and Lung Tissues

Pancreatic and lung tissues were fixed in 10% neutral-buffered formalin for 24 h, washed with tap water, and prepared and stained for light microscopy. For dehydration, serial dilutions of alcohol were applied and thereafter the specimens were cleared in xylene and embedded in paraffin wax in hot air oven at 56 °C for 6 h. Paraffin wax tissue blocks were sectioned by using a microtome at a 5–6 micron thickness. Then, sections were collected on glass slides and deparaffinized. They were stained for routine histological examination using Hematoxylin and Eosin stain (H&E), according to Bancroft and Gamble [71].

### 4.11. Immunohistochemical Examination of iNOS in Pancreatic and Lung Tissues

Paraffin sections were mounted on positively charged slides using the avidin biotin- peroxidase complex (ABC) method. The rabbit iNOS polyclonal antibody was purchased from Gene tex (Cat.No. GTX130246, Dil.: 1:100). Sections from each group were incubated with these antibodies, then the reagents required for the ABC method were added (Vectastain ABC-HRP kit, Vector laboratories, Newark, CA, USA). Marker expression was labeled with peroxidase and colored with diaminobenzidine (DAB, produced by Sigma, Darmstadt, Germany) to detect antigen–antibody complex. Negative controls were included using non-immune serum in place of the primary or secondary antibodies. IHC-stained sections were examined using an Olympus microscope (BX-53). Scoring was performed by the determination of the reaction area percent using ImageJ 1.53K, National Institute of Health, Bethesda, MD, USA.

### 4.12. Statistical Analysis

Data were presented as mean ± SE, and analyzed by one-way ANOVA, followed by the Tukey–Kramer post hoc test in case of normality, and were analyzed using the Kruskal–Wallis test followed by Dunn’s multiple comparisons test in case of non-normality (GraphPad Software, version 9, Boston, MA, USA). The significance limit was set to *p* ≤ 0.05.

## 5. Conclusions

In conclusion, the results of this study suggest that TEP and PS have promising protective benefits in reducing IR-induced damage in both the respiratory system and the pancreas. The proposed mechanism of protection involves reducing the cytokine response and suppressing oxidative stress. Additionally, the mechanism may involve modulating the HMGB1 and IL-22 pathway. These findings may help provide a potential preventive and therapeutic alternative for pancreatic IR injury and associated lung complications. However, further studies are required to explore other possible mechanisms of TEP and PS.

### Limitation of the Current Study

The lack of protein-level validation for gene expression data is one of the main limitations of the current study, along with the incomplete presentation of oxidative stress markers in lung tissue and other parameters. These could be beneficial ideas for future studies of pancreatic IR and its effect on pulmonary tissue.

## Figures and Tables

**Figure 1 ijms-26-02572-f001:**
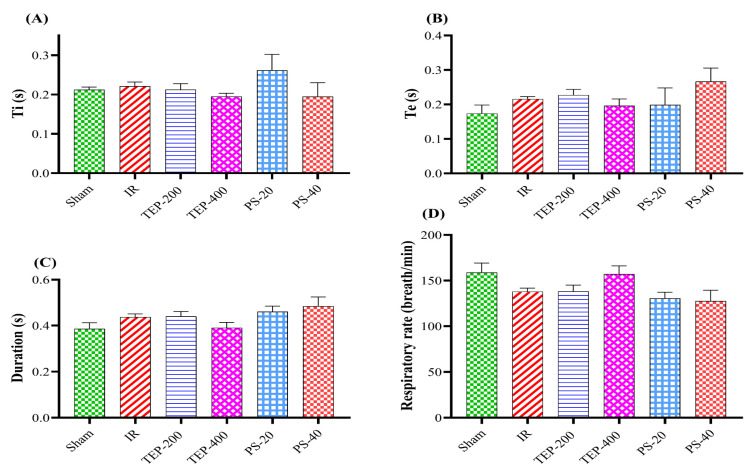
Effect of TEP and PS on the lung function: (**A**) Ti, (**B**) Te, (**C**) duration, and (**D**) RR of pancreatic IR rats. Bars represent mean values of each group ± SEM. Statistical difference is calculated using ANOVA followed by Tukey’s test. No significant difference.

**Figure 2 ijms-26-02572-f002:**
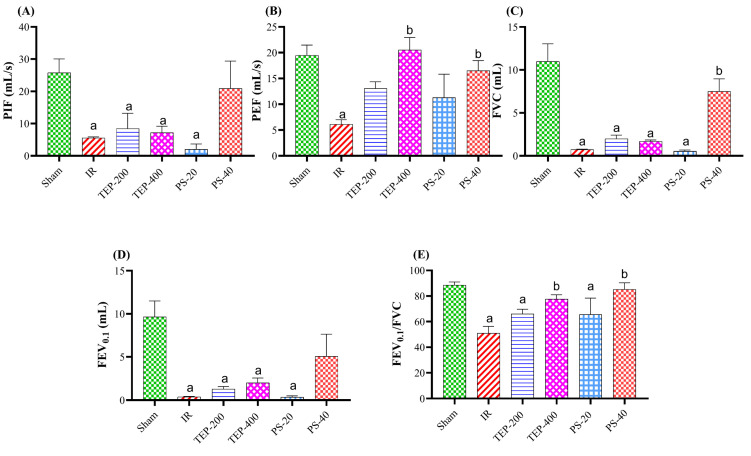
Effect of TEP and PS on the lung functions: (**A**) PIF, (**B**) PEF, (**C**) FVC, (**D**) FEV_0_._1_, and (**E**) FEV_0_._1_/FVC ratio of pancreatic IR rats. Bars represent mean values of each group ± SEM. Statistical difference is calculated using ANOVA followed by Tukey’s test. ^a^ *p* ≤ 0.05 versus sham group, ^b^ *p* ≤ 0.05 versus IR control group.

**Figure 3 ijms-26-02572-f003:**
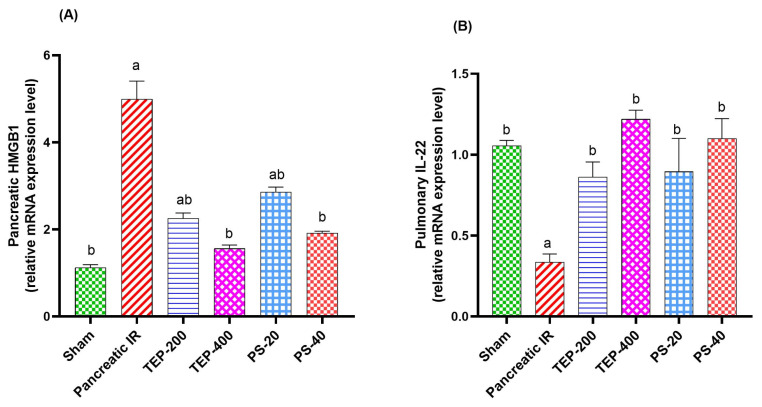
Effects of TEP and PS administration on the expression of high mobility group box 1 protein (HMGB1) and interleukin 22 (IL-22) genes in pancreatic IR-induced injury and associated lung injury by real-time qRT-PCR. (**A**) Pancreatic mRNA expression of HMGB1. (**B**) Lung mRNA expression of IL-22. For all genes, mRNA levels were normalized to glyceraldehyde-3-phosphate dehydrogenase (GAPDH) mRNA level. Bars represent mean values of each group ± SEM. Statistical difference is calculated using ANOVA followed by Tukey’s test. ^a^ *p* ≤ 0.05 versus sham, ^b^ *p* ≤ 0.05 versus pancreatic IR.

**Figure 4 ijms-26-02572-f004:**
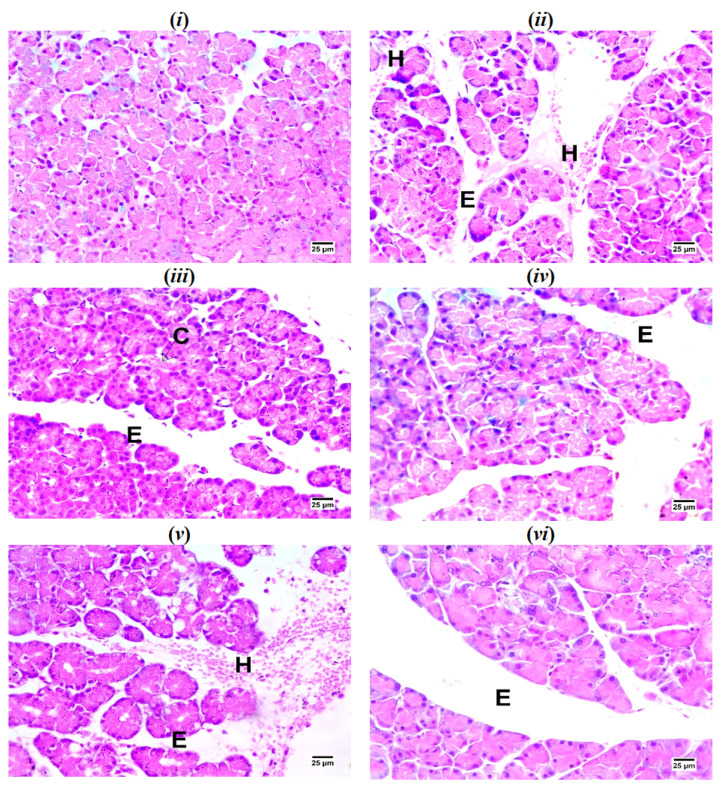
(**i**) Photomicrograph of sham rat showing normal histological structure of pancreas; (**ii**) photomicrograph of IR control showing hemorrhage (H) and edema (E) between pancreatic acini; (**iii**) photomicrograph of TEP-200-treated rat showing congestion of pancreatic blood vessels (C) with edema (E) between pancreatic acini; (**iv**) photomicrograph of TEP-400-treated rat showing edema between pancreatic acini; (**v**) photomicrograph of PS-20-treated rat showing hemorrhage (H) and edema (E) between pancreatic acini; (**vi**) photomicrograph of PS-40-treated rat showing edema (E) between pancreatic acini (H&E).

**Figure 5 ijms-26-02572-f005:**
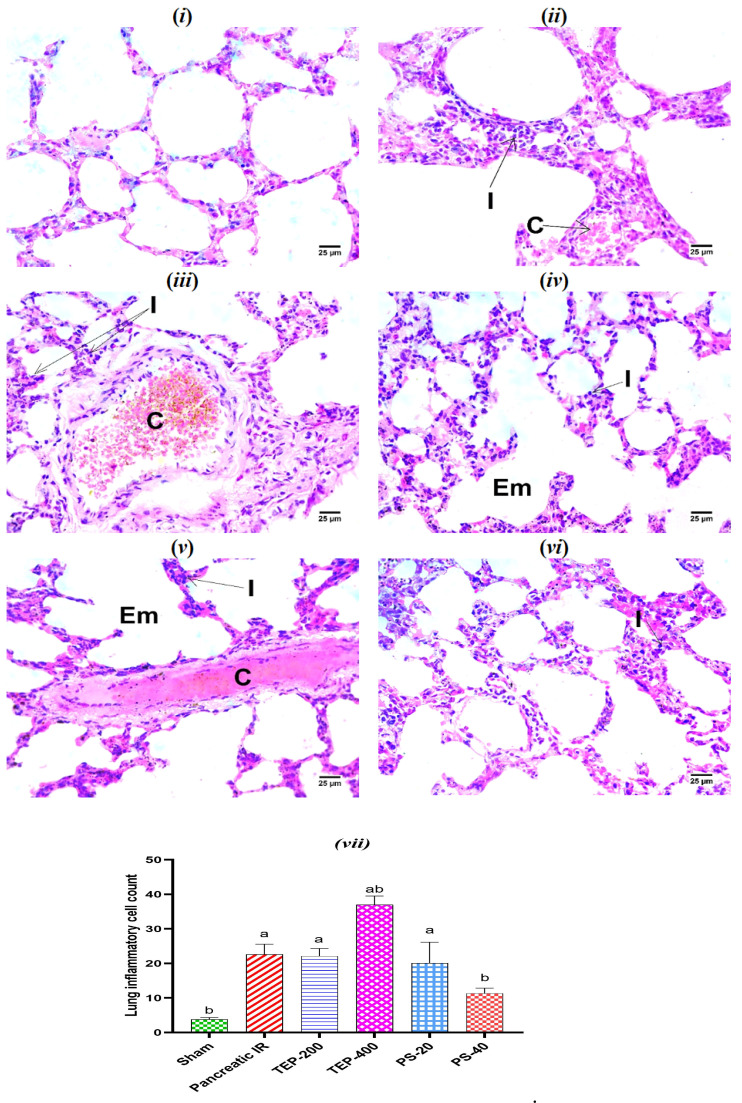
(**i**) Photomicrograph of sham rat showing normal histological structure of alveoli; (**ii**) photomicrograph of IR control rat showing congestion of perialveolar blood vessels (C) with infiltration of interstitial tissue by mononuclear inflammatory cells (I); (**iii**) photomicrograph of TEP-200-treated rat showing congestion of perialveolar blood vessels (C) with infiltration of interstitial tissue by mononuclear inflammatory cells (I); (**iv**) photomicrograph of TEP-400-treated showing emphysematous alveoli (Em) with infiltration of interstitial tissue by mononuclear inflammatory cells (I); (**v**) photomicrograph of PS-20-treated rat showing congestion of perialveolar blood vessels (C) and infiltration of interstitial tissue by mononuclear inflammatory cells (I) with emphysematous alveoli (Em); (**vi**) photomicrograph of PS-40-treated rat showing infiltration of interstitial tissue by low number of mononuclear inflammatory cells (I) (H&E). (**vii**) Bars represent mean values of each group ± SEM. ^a^ *p* ≤ 0.05 versus sham, ^b^ *p* ≤ 0.05 versus pancreatic IR.

**Figure 6 ijms-26-02572-f006:**
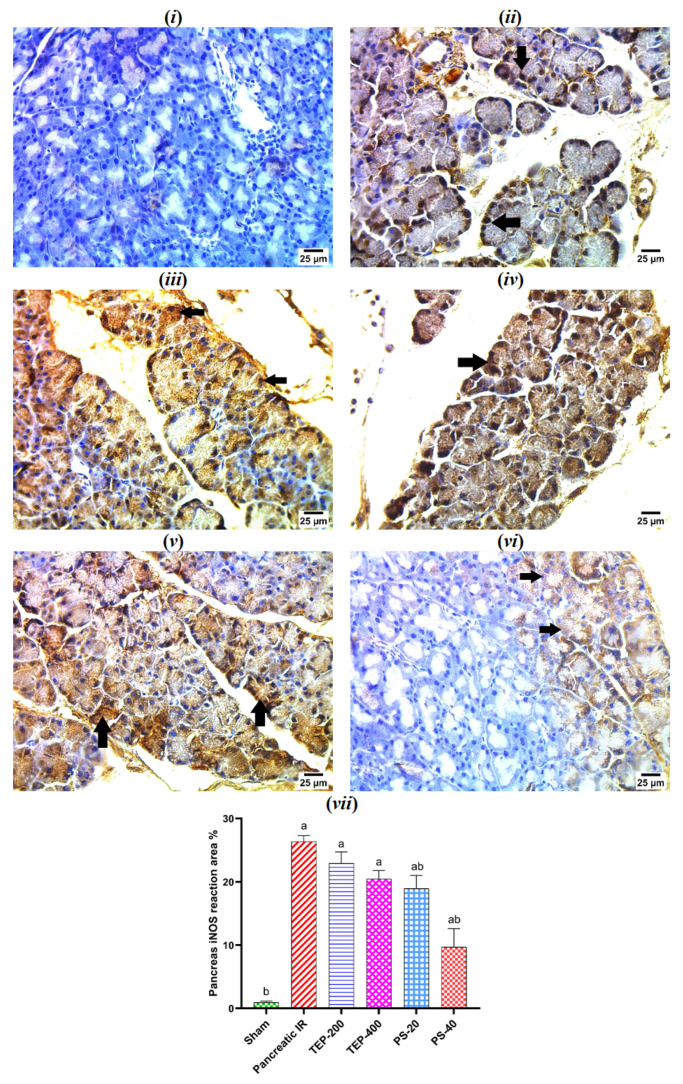
(**i**) Photomicrograph of sham rat showing negative reaction for iNOS in pancreatic acini; (**ii**) photomicrograph of IR control showing strong positive reaction for iNOS in nuclei of pancreatic acini (arrows); (**iii**) photomicrograph of TEP-200-treated rat showing strong positive reaction for iNOS in cytoplasm and nuclei of pancreatic acini (arrows); (**iv**) photomicrograph of TEP-400-treated rat showing strong positive reaction for iNOS in cytoplasm and nuclei of pancreatic acini (arrows); (**v**) photomicrograph of PS-20-treated rat showing strong positive reaction for iNOS in nuclei and cytoplasm of pancreatic acini (arrows); (**vi**) photomicrograph of PS-40-treated rat showing mild positive reaction for iNOS in cytoplasm of some pancreatic acini (arrows) (IHC-peroxidase-DAB). (**vii**) Bars represent mean values of each group ± SEM. ^a^ *p* ≤ 0.05 versus sham, ^b^ *p* ≤ 0.05 versus pancreatic IR.

**Figure 7 ijms-26-02572-f007:**
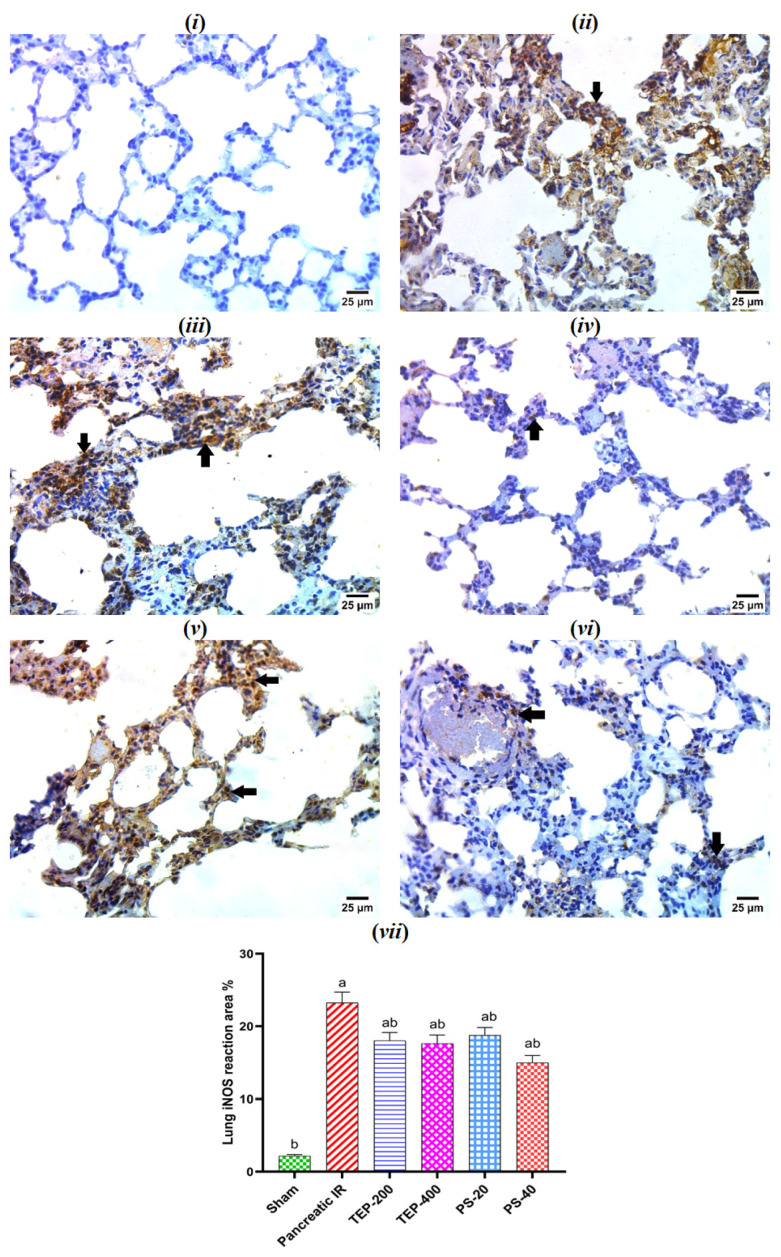
(**i**) Photomicrograph of sham rat showing negative reaction for iNOS in pneumocytes; (**ii**) photomicrograph of IR control rat showing strong positive reaction for iNOS in cytoplasm of pneumocytes; (**iii**) photomicrograph of TEP-200-treated rat showing strong positive reaction for iNOS in cytoplasm and nuclei of pneumocytes (arrows); (**iv**) photomicrograph of TEP-400-treated rat showing very mild positive reaction for iNOS in cytoplasm of few numbers of pneumocytes (arrows); (**v**) photomicrograph of PS-20-treated rat showing strong positive reaction for iNOS in cytoplasm and nuclei of pneumocytes (arrows); (**vi**) photomicrograph of PS-40-treated rat showing very mild positive reaction for iNOS in cytoplasm of pneumocytes (arrows) (IHC-peroxidase-DAB). (**vii**) Bars represent mean values of each group ± SEM. ^a^ *p* ≤ 0.05 versus sham, ^b^ *p* ≤ 0.05 versus pancreatic IR.

**Table 1 ijms-26-02572-t001:** Effect of TEP and PS on the serum pancreatic enzymes of pancreatic IR rats.

Group	Amylase (ng/mL)	Lipase (pg/mL)	TAP (ng/mL)
Sham	5.0 ^b^ ± 0.46	32.4 ^b^ ± 1.57	0.4 ^b^ ± 0.04
IR control	21.9 ^a^ ± 1.11	144.4 ^a^ ± 6.23	3.1 ^a^ ± 0.09
TEP-200	14.5 ^ab^ ± 0.66	91.4 ^ab^ ± 4.64	2.0 ^ab^ ± 0.05
TEP-400	7.6 ^b^ ± 0.44	45.5 ^b^ ± 1.85	1.0 ^ab^ ± 0.07
PS-20	14.1 ^ab^ ± 0.55	90.4 ^ab^ ± 4.15	2.2 ^ab^ ± 0.10
PS-40	7.3 ^b^ ± 0.40	48.1 ^b^ ± 1.87	1.4 ^ab^ ± 0.06

Results are presented as mean ± SEM (n = 6). Statistical difference is calculated using ANOVA followed by Tukey’s test. ^a^ Significantly different in comparison with sham group (*p* ≤ 0.05). ^b^ Significantly different in comparison with IR control group (*p* ≤ 0.05).

**Table 2 ijms-26-02572-t002:** Effect of TEP and PS on the pancreatic inflammatory markers of pancreatic IR rats.

Group	Pancreatic TNF-α (pg/mg Protein)	Pancreatic IL-1β (pg/mg Protein)	Pancreatic NF-κB (ng/mg Protein)
Sham	30.3 ^b^ ± 1.00	40.5 ^b^ ± 1.56	29.6 ^b^ ± 1.18
IR control	324.5 ^a^ ± 5.37	212.9 ^a^ ± 4.71	287.7 ^a^ ± 15.34
TEP-200	194.6 ^ab^ ± 2.47	163.3 ^ab^ ± 3.98	156.1 ^ab^ ± 9.85
TEP-400	72.2 ^ab^ ± 6.50	56.5 ^b^ ± 4.83	72.9 ^ab^ ± 2.45
PS-20	248.6 ^ab^ ± 11.03	159.5 ^ab^ ± 8.22	155.6 ^ab^ ± 3.44
PS-40	100.5 ^ab^ ± 7.22	59.2 ^b^ ± 3.60	73.6 ^ab^ ± 3.84

Results are presented as mean ± SEM (n = 6). ^a^ Significantly different in comparison with sham group (*p* ≤ 0.05). ^b^ Significantly different in comparison with IR control group (*p* ≤ 0.05).

**Table 3 ijms-26-02572-t003:** Effect of TEP and PS on the pulmonary inflammatory markers of pancreatic IR rats.

Group	Lung TNF-α (pg/mg Protein)	Lung IL-1β (pg/mg Protein)	Lung NF-κB (ng/mg Protein)
Sham	33.5 ^b^ ± 2.77	31.5 ^b^ ± 1.03	18.4 ^b^ ± 0.95
IR control	177.3 ^a^ ± 5.48	205.6 ^a^ ± 3.88	90.6 ^a^ ± 3.47
TEP-200	72.1 ^ab^ ± 4.39	136.6 ^ab^ ± 3.43	50.9 ^ab^ ± 2.18
TEP-400	31.2 ^b^ ± 1.28	51.7 ^ab^ ± 4.11	33.1 ^ab^ ± 2.77
PS-20	96.0 ^ab^ ± 4.98	148.6 ^ab^ ± 3.73	62.0 ^ab^ ± 2.12
PS-40	70.3 ^ab^ ± 2.67	53.8 ^ab^ ± 3.69	34.3 ^ab^ ± 1.71

Results are presented as mean ± SEM (n = 6). Statistical difference is calculated using ANOVA followed by Tukey’s test. ^a^ Significantly different in comparison with sham group (*p* ≤ 0.05). ^b^ Significantly different in comparison with IR control group (*p* ≤ 0.05).

**Table 4 ijms-26-02572-t004:** Effect of TEP and PS on the pancreatic oxidation markers of pancreatic IR rats.

Group	Pancreatic MDA (nmol/mg Protein)	Pancreatic GPx (nmol/mg Protein)	Pancreatic MPO (ng/mg Protein)
Sham	0.3 ^b^ ± 0.01	3.0 ^b^ ± 0.23	0.9 ^b^ ± 0.09
IR control	1.8 ^a^ ± 0.03	0.7 ^a^ ± 0.03	6.4 ^a^ ± 0.31
TEP-200	1.1 ^ab^ ± 0.05	1.9 ^ab^ ± 0.12	4.9 ^ab^ ± 0.18
TEP-400	0.7 ^ab^ ± 0.03	3.0 ^b^ ± 0.09	2.9 ^ab^ ± 0.08
PS-20	1.3 ^ab^ ± 0.05	1.9 ^ab^ ± 0.09	5.2 ^ab^ ± 0.27
PS-40	0.8 ^ab^ ± 0.03	2.6 ^b^ ± 0.11	3.0 ^ab^ ± 0.23

Results are presented as mean ± SEM (n = 6). Statistical difference is calculated using ANOVA followed by Tukey’s test. ^a^ Significantly different in comparison with sham group (*p* ≤ 0.05). ^b^ Significantly different in comparison with IR control group (*p* ≤ 0.05).

## Data Availability

The original contributions presented in this study are included in the article. Further inquiries can be directed to the corresponding author.

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
