# Peer review of "Tephrosia purpurea*, with (-)-Pseudosemiglabrin as the Major Constituent, Alleviates Severe Acute Pancreatitis-Mediated Acute Lung Injury by Modulating HMGB1 and IL-22"

_ijms, 2025, doi:10.3390/ijms26062572_

Round 1
Reviewer 1 Report
Comments and Suggestions for Authors
The aim of the research conducted by Gamal A. Soliman and colleagues is to investigate the role of Tephrosia purpurea (TEP) and its active ingredient pseudosemiglazrin (PS) in alleviating severe acute pancreatitis and associated acute lung injury. In my opinion, the authors correctly planned and performed the experiments, using various research techniques. The results of the research are described carefully and clearly, understandably for the reader, correctly interpreting the obtained data. The introduction contains basic information about the subject of the research, which introduces the reader to the issues related to the conducted experiments. The discussion comprehensively compares the obtained data with the published results of other researchers. The conclusions were correctly formulated taking into account the obtained results. The literature was correctly selected.
I ask the authors to include information in the manuscript in the materials and methods section why they administered the extract and pseudosemiglazrin to laboratory animals in these specific doses. What influenced the decision to administer pseudosemiglabrin at a dose of 20 and 40 mg/kg, and T.purpurea at a dose of 200 and 400 mg/kg.
Author Response
We thank the reviewer for the nice comments and valuable points for imporovements.
Two references have been added to the manuscript to support why the extract and pseudosemiglabrin were given to laboratory animals at these specific doses
- purpurea (200 and 400 mg/kg).
Hussain T, Fareed S, Siddiqui HH, Vijaykumar M, Rao CV. Acute and subacute oral toxicity evaluation of Tephrosia purpurea extract in rodents. Asian Pacific journal of tropical disease. 2012 Apr 1;2(2):129-32.
- Pseudosemiglabrin (20 and 40 mg/kg)
Balaha MF, Alamer AA, Abdel-Kader MS, Alharthy KM. Ameliorative potential of (-) pseudosemiglabrin in mice with pilocarpine-induced epilepsy: antioxidant, anti-inflammatory, anti-apoptotic, and neurotransmission modulation. International Journal of Molecular Sciences. 2023 Jun 28;24(13):10773.
Reviewer 2 Report
Comments and Suggestions for Authors
In the study the role of Tephrosia purpurea (TEP) and its active constituent pseudosemiglabrin (PS) in alleviating severe acute pancreatitis and its associated acute lung injury have been investigated.
I have some questions and comments:
1. Your data presented in Table 4 shows that pancreatic MPO activities in IR control group were reduced in comparison to sham rats. Are presented data correct? If yes, how could be explained this reduction of pancreatic MPO activities in IR control rats? Myeloperoxidase is enzyme which is known as an important mediator of oxidative stress and inflammation (has pro-oxidative and proinflammatory properties).
Moreover, in Discussion (lines 362-364) you wrote: “In this study, pancreatic IR resulted in severe oxidative stress characterized by increased free radical products MDA and MPO activity as well as decreased GPx activity in pancreatic tissue of IR control rats.” This statement about MPO activities is not in agreement with presented data.
2. In Abstract you wrote (lines 27-29): “Additionally, TEP and PS showed protective effects on pancreatic and lung tissues by alleviating the pathological damage, reducing serum levels of trypsinogen activation peptide (TAP), lipase, and amylase, decreasing oxidative stress markers MDA, GPx, and MPO, …”
Protective effects of TEP and PS were not associated with decreasing GPx. In addition, glutathione peroxidase is an antioxidant enzyme and is not marker of oxidative stress.
3. Why are not presented data showing the effects of pancreatic IR and treatment with TEP or PS on MDA, MPO, and GPx in lung tissue? The data dealing with inflammatory markers were presented for both pancreas ang lung tissue (Tables 2 and 3).
4. Were the changes in gene expression (mRNA) for HMGB1 and IL-22 associated with corresponding changes in protein levels? These data are missing.
I am asking because the changes in gene expression not always reflect the changes at the protein levels. And proteins are biomolecules with crucial role in realization of cellular functions.
5. Title of your paper is “Tephrosia purpurea and (-)-pseudosemiglabrin the major constituent; alleviate severe acute pancreatitis-mediated acute lung injury by modulating HMGB1 and IL-22”. You showed at the mRNA levels the effects of TEP and PS on IL-22 in lung tissue but not in pancreas. On the other hand, for HMGB1 are presented pancreatic but not pulmonary relative mRNA expression levels. What was reason for these differences by examination and presentation of HMGB1 and IL-22?
6. You presented data showing Th22 gene expression. Is it correct? Th22 are cells and their function is also production of interleukin-22.
7. In lines 152-153 you wrote: “Similarly, pancreatic MPO was replenished by 3.1, 5.3, 3.2, or 5.6% compared to the IR control group, respectively.” It is not clear what means %.
Author Response
Comments and Suggestions for Authors
In the study the role of Tephrosia purpurea (TEP) and its active constituent pseudosemiglabrin (PS) in alleviating severe acute pancreatitis and its associated acute lung injury have been investigated.
I have some questions and comments:
- Your data presented in Table 4 shows that pancreatic MPO activities in IR control group were reduced in comparison to sham rats. Are presented data correct? If yes, how could be explained this reduction of pancreatic MPO activities in IR control rats? Myeloperoxidase is enzyme which is known as an important mediator of oxidative stress and inflammation (has pro-oxidative and proinflammatory properties). Moreover, in Discussion (lines 362-364) you wrote: “In this study, pancreatic IR resulted in severe oxidative stress characterized by increased free radical products MDA and MPO activity as well as decreased GPx activity in pancreatic tissue of IR control rats.” This statement about MPO activities is not in agreement with presented data.
Response:
We appreciate the reviewer's note regarding the difference in the data and the discussion. While reviewing the raw data for Table 4, we found that IR values were mistakenly replaced with sham values during the finalization of the manuscript; it was simply a copy-paste error. We have reviewed the entire manuscript data and corrected the error.
- In Abstract you wrote (lines 27-29): “Additionally, TEP and PS showed protective effects on pancreatic and lung tissues by alleviating the pathological damage, reducing serum levels of trypsinogen activation peptide (TAP), lipase, and amylase, decreasing oxidative stress markers MDA, GPx, and MPO, …”
Protective effects of TEP and PS were not associated with decreasing GPx. In addition, glutathione peroxidase is an antioxidant enzyme and is not marker of oxidative stress.
Response:
We thank the reviewer for pointing out this inconsistency. We agree that GPx is an antioxidant enzyme and not a direct marker of oxidative stress. The protective effects of TEP and PS were associated with restoration of GPx activity rather than its reduction. We have modified the abstract to accurately reflect this, as follows:
“Additionally, TEP and PS showed protective effects on pancreatic and lung tissues by alleviating the pathological damage, reducing serum levels of trypsinogen activation peptide (TAP), lipase, and amylase, decreasing oxidative stress markers such as MDA and MPO, and restoring antioxidant enzyme activity (GPx).”
- Why are not presented data showing the effects of pancreatic IR and treatment with TEP or PS on MDA, MPO, and GPx in lung tissue? The data dealing with inflammatory markers were presented for both pancreas ang lung tissue (Tables 2 and 3).
Response:
We thank the reviewer for the valuable advice on adding MDA, MPO and GPx measurements in lung tissues. In this article the main endpoints were pro-inflammatory mediators in the lung, however markers of oxidative stress such as MDA, MPO and GPx are equally important for understanding TEP and PS activity. We therefore add this to the study limitations.
- Were the changes in gene expression (mRNA) for HMGB1 and IL-22 associated with corresponding changes in protein levels? These data are missing.
I am asking because the changes in gene expression not always reflect the changes at the protein levels. And proteins are biomolecules with crucial role in realization of cellular functions.
Response:
We fully acknowledge the reviewer's concerns. We fully agree that the relationship between mRNA and protein levels may or may not be correlated. Protein data are essential for understanding functional implications. In this study, we chose to look at mRNA levels as a starting point for assessing the effect of TEP and PS; however, we agree and appreciate that protein level validation is of paramount importance. Therefore, we have included this part within the limitations of the current study
- Title of your paper is “Tephrosia purpurea and (-)-pseudosemiglabrin the major constituent; alleviate severe acute pancreatitis-mediated acute lung injury by modulating HMGB1 and IL-22”. You showed at the mRNA levels the effects of TEP and PS on IL-22 in lung tissue but not in pancreas. On the other hand, for HMGB1 are presented pancreatic but not pulmonary relative mRNA expression levels. What was reason for these differences by examination and presentation of HMGB1 and IL-22?
Response:
Regarding the differential presentation of HMGB1 and IL-22 data:
The measurements of IL-22 in the lung tissue and HMGB1 in the pancreatic tissue were based on their respective roles in the pathophysiology of acute lung injury and acute pancreatitis. IL-22 is mostly involved in lung tissue repair and immune modulation [1,2], whereas HMGB1 plays an acute role in pancreatic inflammation and injury [3,4]. However, we agree that presenting data for both markers in both tissues would provide a more balanced and comprehensive analysis. Therefore, we included this part to the limitation of the current study.
[1] Wu, Z., Hu, Z., Cai, X. et al. Interleukin 22 attenuated angiotensin II induced acute lung injury through inhibiting the apoptosis of pulmonary microvascular endothelial cells. Sci Rep 7, 2210 (2017). https://doi.org/10.1038/s41598-017-02056-w.
[2] Taghavi S, Jackson-Weaver O, Abdullah S, Wanek A, Drury R, Packer J, Cotton-Betteridge A, Duchesne J, Pociask D, Kolls J. Interleukin-22 mitigates acute respiratory distress syndrome (ARDS). PLoS One. 2021 Oct 1;16(10):e0254985. doi: 10.1371/journal.pone.0254985.
[3] Kang R, Zhang Q, Hou W, Yan Z, Chen R, Bonaroti J, Bansal P, Billiar TR, Tsung A, Wang Q, Bartlett DL, Whitcomb DC, Chang EB, Zhu X, Wang H, Lu B, Tracey KJ, Cao L, Fan XG, Lotze MT, Zeh HJ 3rd, Tang D. Intracellular Hmgb1 inhibits inflammatory nucleosome release and limits acute pancreatitis in mice. Gastroenterology. 2014 Apr;146(4):1097-107. doi: 10.1053/j.gastro.2013.12.015. Epub 2013 Dec 17. PMID: 24361123.
[4] Shen X, Li WQ. High-mobility group box 1 protein and its role in severe acute pancreatitis. World J Gastroenterol. 2015 Feb 7;21(5):1424-35. doi: 10.3748/wjg.v21.i5.1424.
- You presented data showing Th22 gene expression. Is it correct? Th22 are cells and their function is also production of interleukin-22.
Response:
We thank the reviewer for highlighting this point, which was a terminology error. The revised text now accurately distinguishes between Th22 cells (the cellular source) and IL-22 (the cytokine they produce), ensuring proper mechanistic interpretation.
- In lines 152-153 you wrote: “Similarly, pancreatic MPO was replenished by 3.1, 5.3, 3.2, or 5.6% compared to the IR control group, respectively.” It is not clear what means %.
Response:
Corrected.
Reviewer 3 Report
Comments and Suggestions for Authors
- A. Soliman et al. evaluated the putative protective role of the legume species Tephrosia purpurea (TEP) and its active metabolite pseudosemiglabrin (PS) in the context of pancreatic ischemia-reperfusion injury (IR) and associated acute lung injury in an experimental rat model. The animals were treated with two different concentrations of the extracts (TEP: 200 mg/kg and 400 mg/kg; PS: 20 mg/kg and 40 mg/kg. The authors found partially improved lung function parameters as well as protective effects on pancreatic and lung tissue, which they were able to demonstrate correlatively on the basis of morphological and biochemical parameters. In summary, the authors conclude that treatment with TEP and PS alleviates IR of the pancreas and lungs by reducing oxidative stress and improving inflammatory processes, from which they consequently derive a therapeutic option.
This is a well-understood and systematically conducted study. The authors first describe the clinical and pathophysiological basis as well as the previously insufficiently understood pharmacological or possible phythotherapeutic effect of the extracts, which was the reason for conducting this analysis. The technical-experimental approach was an experimental rat model, whereby the authors used two different concentrations of the substances as well as corresponding positive (IR) and negative (Sham) controls. The results are presented in an easily understandable form, including systematically performed histo-morphological, functional and biochemical evaluations. In the subsequent discussion, the authors demonstrate the clinical relevance of their results and interpret the individual cellular effects in the patho-physiological context.
Basically, I have only a minor critical comment on the data set presented here.
- So far in their discussion, the authors have outlined the basic pathophysiological aspects of IR, but have described purely correlatively how the treatments have affected functional parameters, oxidative stress, markers of humoral and cellular inflammation in both organs. The exact mechanism remains completely unclear. A more in-depth discussion of this would be desirable, possibly with reference to other studies.
- As far as I have seen, there is no scaling in the histomorphological analyses.
- Ad Figure 5: In the histomorphological assessment, quantifications in group comparison would be useful (e.g. of the inflammatory reaction).
- The quantifications in Fig. 6/7 are also unclear to me because no overview colorations are shown.
- I recommend an indication of the statistical tests performed in the legend under each figure, which has not yet been implemented stringently, and the use of boxplots instead of bar graphs, in accordance with modern standards.
Author Response
Comments and Suggestions for Authors
- Soliman et al. evaluated the putative protective role of the legume species Tephrosia purpurea (TEP) and its active metabolite pseudosemiglabrin (PS) in the context of pancreatic ischemia-reperfusion injury (IR) and associated acute lung injury in an experimental rat model. The animals were treated with two different concentrations of the extracts (TEP: 200 mg/kg and 400 mg/kg; PS: 20 mg/kg and 40 mg/kg. The authors found partially improved lung function parameters as well as protective effects on pancreatic and lung tissue, which they were able to demonstrate correlatively on the basis of morphological and biochemical parameters. In summary, the authors conclude that treatment with TEP and PS alleviates IR of the pancreas and lungs by reducing oxidative stress and improving inflammatory processes, from which they consequently derive a therapeutic option.
This is a well-understood and systematically conducted study. The authors first describe the clinical and pathophysiological basis as well as the previously insufficiently understood pharmacological or possible phythotherapeutic effect of the extracts, which was the reason for conducting this analysis. The technical-experimental approach was an experimental rat model, whereby the authors used two different concentrations of the substances as well as corresponding positive (IR) and negative (Sham) controls. The results are presented in an easily understandable form, including systematically performed histo-morphological, functional and biochemical evaluations. In the subsequent discussion, the authors demonstrate the clinical relevance of their results and interpret the individual cellular effects in the patho-physiological context.
Basically, I have only a minor critical comment on the data set presented here.
- So far in their discussion, the authors have outlined the basic pathophysiological aspects of IR, but have described purely correlatively how the treatments have affected functional parameters, oxidative stress, markers of humoral and cellular inflammation in both organs. The exact mechanism remains completely unclear. A more in-depth discussion of this would be desirable, possibly with reference to other studies.
Response:
We sincerely thank the reviewer for his/her insightful feedback and agree that a deeper exploration of the mechanistic underpinnings of our findings would strengthen the discussion.
Discussion has been enhanced and revised
- As far as I have seen, there is no scaling in the histomorphological analyses.
Response:
All histopathological figures are provided at 25 μm scale and is shown within each image in the right corner.
- Ad Figure 5: In the histomorphological assessment, quantifications in group comparison would be useful (e.g. of the inflammatory reaction).
Response:
Done.
- The quantifications in Fig. 6/7 are also unclear to me because no overview colorations are shown.
Response:
Immunohistochemical scoring is already done and incorporated as part of figure 6&7 (bar graph no vii).
- I recommend an indication of the statistical tests performed in the legend under each figure, which has not yet been implemented stringently, and the use of boxplots instead of bar graphs, in accordance with modern standards.
Response:
Done.
Round 2
Reviewer 2 Report
Comments and Suggestions for Authors
Comments to responses of authors
Comment to authors response 1:
In revised manuscript you replaced in Table 4 also values for TEP-200 with TEP-400 and values for TS-20 with TS-40.
Moreover, in revised manuscript is the following information:”…MPO increased by 86% in the pancreatic tissue of the IR control group compared to sham rats.” (lines 154-155). The information (86%) is in revised version of manuscript not correct (based on data presented in Table 4).
Comment to authors response 4:
The protein data are really essential for understanding functional implications and are missing in presented data.
Comment to authors response 5:
HMGB1 plays an important role also in pathophysiology of acute lung injury. Involved could be increased circulating levels of this proinflammatory cytokine as well as its increased lung production in consequence of enhanced oxidative stress in lung and increased generation of reactive oxygen or nitrogen species.
The data presented in your study do not document that the effects of TEP and PS on acute lung injury involve directly the modulation of HMGB1.
Comment to authors response 6:
I asked if was correct the presentation of data showing Th22 gene expression.
Th22 are cells and not simple protein encoded by specific mRNA. Exists specific primer for mRNA encoding Th22 cells?
Author Response
Comment to author’s response 1:
In revised manuscript you replaced in Table 4 also values for TEP-200 with TEP-400 and values for TS-20 with TS-40.
Moreover, in revised manuscript is the following information:”…MPO increased by 86% in the pancreatic tissue of the IR control group compared to sham rats.” (lines 154-155). The information (86%) is in revised version of manuscript not correct (based on data presented in Table 4).
Response: as mentioned in the previous revision and while going through the raw data for Table 4, we corrected the MPO data for the mistakenly placed values in the draft manuscript. And thanks for your observation for the calculated percentage, correction done.
Comment to author’s response 4:
The protein data are really essential for understanding functional implications and are missing in presented data.
Response: We do agree with this point, and as we mentioned before, that we chose to look at mRNA levels as a starting point to evaluate the effect of TEP and PS. Moreover, in support to our finding:
- Inducible nitric oxide synthase (iNOS), which is downstream of HMGB1 in inflammatory signaling pathways.
- HMGB1 is a damage-associated molecular pattern (DAMP) that is released by necrotic cells or secreted by immune cells in response to stress or infection.
- Activation of TLR4 and RAGE: HMGB1 binds to Toll-like receptor 4 (TLR4) and Receptor for Advanced Glycation End-products (RAGE), triggering intracellular signaling cascades.
- NF-κB Pathway Activation: These receptors activate the NF-κB pathway, which leads to the transcriptional upregulation of pro-inflammatory genes, including iNOS.
- iNOS Induction: Once activated, NF-κB enhances the expression of NOS2, the gene encoding iNOS, leading to the production of nitric oxide (NO), a key molecule in inflammation and immune responses.
Comment to author’s response 5:
HMGB1 plays an important role also in pathophysiology of acute lung injury. Involved could be increased circulating levels of this proinflammatory cytokine as well as its increased lung production in consequence of enhanced oxidative stress in lung and increased generation of reactive oxygen or nitrogen species.
The data presented in your study do not document that the effects of TEP and PS on acute lung injury involve directly the modulation of HMGB1.
Response:
As we mentioned in the previous comment, we suggest that regulation of HMGB1 mRNA by TEP and PS could be involved in mechanism involved in their anti-inflammatory activities against acute pancreatitis-mediated acute lung injury. Moreover, in support to our finding:
- Inducible nitric oxide synthase (iNOS), which is downstream of HMGB1 in inflammatory signaling pathways.
- HMGB1 is a damage-associated molecular pattern (DAMP) that is released by necrotic cells or secreted by immune cells in response to stress or infection.
- Activation of TLR4 and RAGE: HMGB1 binds to Toll-like receptor 4 (TLR4) and Receptor for Advanced Glycation End-products (RAGE), triggering intracellular signaling cascades.
- NF-κB Pathway Activation: These receptors activate the NF-κB pathway, which leads to the transcriptional upregulation of pro-inflammatory genes, including iNOS.
- iNOS Induction: Once activated, NF-κB enhances the expression of NOS2, the gene encoding iNOS, leading to the production of nitric oxide (NO), a key molecule in inflammation and immune responses.
Comment to author’s response 6:
I asked if was correct the presentation of data showing Th22 gene expression.
Th22 are cells and not simple protein encoded by specific mRNA. Exists specific primer for mRNA encoding Th22 cells?
Response:
We thank the reviewer for this observation. We do agree with this point, Th22 cells are a subset of T cells characterized by their distinct cytokine profile in which IL-22 is the main effector molecule of Th22 and a key regulators of epithelial barrier function and inflammation.
In the current study, we used the following primer sequence (forward 5-GGGCTTCCAGGGTGCTTCGC-3′, reverse 5-CCTCAGTTCACCGAGAACCCCA-3’ (NM_201270.1) according to Huai et al. (2012) who used this primer set to analyze the gene expression of Th22. By checking this sequence on Primer-BLAST (https://www.ncbi.nlm.nih.gov/tools/primer-blast/index.cgi?GROUP_TARGET=on) we found it incorrect.
The authors apologize for this error and appreciate the reviewer’s insightful comment. So, we decided to exclude this from our research. This change didn’t affect the main findings or the study's overall conclusion.
Reference:
Huai JP, Sun XC, Chen MJ, Jin Y, Ye XH, Wu JS, Huang ZM. Melatonin attenuates acute pancreatitis-associated lung injury in rats by modulating interleukin 22. World J Gastroenterol. 2012 Sep 28;18(36):5122-8. doi: 10.3748/wjg.v18.i36.5122. PMID: 23049224; PMCID: PMC3460342.
Reviewer 3 Report
Comments and Suggestions for Authors
The authors have responded sufficiently to my constructive criticism. In their detailed response, they have satisfactorily optimized the illustrations/figure legends I criticized, mapped the changes transparently, and thus dispelled some of my doubts. Overall, the manuscript in its current form is more scientifically valuable, also considering formal improvements regarding the discussion and references. I have no objections to the validity of the study.
Regarding the ANOVA in Figure 1: If a Tukey test was performed post hoc in Figure 1, please indicate this in the legend.
Author Response
Comments and Suggestions for Authors
The authors have responded sufficiently to my constructive criticism. In their detailed response, they have satisfactorily optimized the illustrations/figure legends I criticized, mapped the changes transparently, and thus dispelled some of my doubts. Overall, the manuscript in its current form is more scientifically valuable, also considering formal improvements regarding the discussion and references. I have no objections to the validity of the study.
Regarding the ANOVA in Figure 1: If a Tukey test was performed post hoc in Figure 1, please indicate this in the legend.
Response: done.
Round 3
Reviewer 2 Report
Comments and Suggestions for Authors
Authors have done corresponding correction in MPO data. In revised manuscript were also excluded incorrect data dealing with Th22 cells.
I accept explanations of authors to comments 4 and 5.